# Computer-Aided Rehabilitation Supported by Zygomatic Implants: A Cohort Study Comparing Atrophic with Oncologic Patients after Five Years of Follow-Up

**DOI:** 10.3390/jcm9103254

**Published:** 2020-10-12

**Authors:** Gerardo Pellegrino, Francesco Basile, Daniela Relics, Agnese Ferri, Francesco Grande, Achille Tarsitano, Claudio Marchetti

**Affiliations:** 1Oral and Maxillofacial Surgery Unit, DIBINEM, University of Bologna, 40100 Bologna, Italy; gerardo.pellegrino2@unibo.it (G.P.); basile@dottorbasile.it (F.B.); danielarelics2001@yahoo.it (D.R.); agnese.ferri3@unibo.it (A.F.); francesco.grande6@unibo.it (F.G.); 2Oral and Maxillofacial Surgery Unit, Azienda Ospedaliero-Universitaria di Bologna, DIBINEM, University of Bologna, 40100 Bologna, Italy; claudio.marchetti@unibo.it

**Keywords:** zygomatic implants, dental implants, atrophic maxilla, oncologic patients, computer-aided planning, computer-assisted surgery, oral rehabilitation

## Abstract

The aim of this study was to evaluate the survival and clinical success rate, complications, and patients’ quality of life after computer-aided rehabilitation supported by zygomatic implants in cases of severe maxillary atrophy (ten patients) and in bone defects in oncologic patients (ten patients). All patients underwent computer-aided planning and surgery. Seventy-three zygomatic implants were placed. The mean follow-up period was 39.9 months. Implant survival and clinical success rate, the effectiveness of planning the implant length, biological and prosthetic complications, and the quality of life were evaluated. The five-year implant survival rate for patients with maxillary atrophy and oncologic patients was 97.4% and 96.7%, respectively. The prosthetic survival rate was 100%. Two implant failures occurred in the first year. One implant failure was observed in each group. Minor biological and prosthetic complications occurred in both groups without significant differences. All complications were managed without affecting the treatment. The quality of life increased by 71.3% in the atrophic group and by 82.9% in the oncologic group. Zygomatic implant rehabilitation seems to be a reliable technique for patients with maxillary atrophy and for oncologic patients. The three-dimensional computer-aided approach allows the surgeon to plan the surgery and increase its predictability. Early prosthesis loading certainly allows for better functional outcomes.

## 1. Introduction

Zygomatic implant-supported rehabilitation constitutes a reliable alternative to reconstructive procedures for the restoration of atrophic edentulous maxillae as well as for maxillectomy defects [1,2,3,4,5]. The need to achieve stability of the prostheses and to reduce the invasiveness and the duration of the surgical procedures are the main reasons for the advent of zygomatic implants [5,6]. Bone grafting may be contraindicated due to general health conditions or after previous grafting failure in some cases. In addition, reconstructive procedures require prolonged treatment and healing time. Despite the widespread use of biomaterials, the treatment of severely atrophic maxillae can often require harvesting from intraoral or extraoral donor sites [7] with increased patient morbidity and additional financial burden [8].

Rehabilitation with zygomatic implants in severely atrophic maxillae has been proposed as an alternative to reconstructive procedures. The treatment involves the positioning of two zygomatic implants in the posterior area of the maxilla and two to four standard dental implants in the anterior maxilla or four zygomatic implants for supporting a full-arch prosthesis [9,10]. Many studies where zygomatic implants were inserted alone or in combination with conventional implants in severely atrophic maxillae have reported high survival rates comparable to conventional implants [11,12,13,14].

The high success rate reported in patients with atrophic maxillae has promoted the use of this technique to rehabilitate patients who have undergone maxillary resection for oncologic issues [15].

The rehabilitation of these patients is often quite challenging for the surgeon and requires a high degree of surgical skill. Some interesting classifications have been suggested to standardize the decision-making process for surgical planning, prosthetic solutions, and dental rehabilitation [16,17]. Nonetheless, the anatomical defect after resection for cancer remains unique.

The literature about zygomatic implants in oncologic patients is scarce compared to that in patients with atrophic maxillae. Usually, studies that evaluate zygomatic implants for treating patients after maxillary resection report a success rate considerably lower than studies about zygomatic implants in patients with atrophic maxillae. In a systematic review on survival and complications of zygomatic implants in patients with atrophic maxillae and in oncologic patients, Chrcanovic et al. found that the cumulative survival rate over a 12-year period was 96.7%. The zygomatic implant survival rate for patients with atrophic maxillae ranges from 95.8% to 100%, while that for oncologic patients ranges from 78.6% to 91.7% [2].

Zygomatic implant rehabilitation in patients with maxillary atrophy and in oncologic patients has certainly gained popularity as a new treatment option for more rapid restoration with a reduced number of surgical steps, complications, and costs. However, zygomatic implant placement is considered a challenging procedure due to the angled trajectory and implant length [18,19]. The limited intraoperative visibility of the deeper part of the malar bone and the close proximity of anatomical structures can be challenging. This issue could be overcome by using computer-assisted planning and implant placement to achieve safe and clinically acceptable implant positioning [20,21].

Although studies have been published regarding zygomatic implants in patients with atrophic maxillae and in oncologic patients, comparative studies between these groups have not been reported to the best of our knowledge.

The aim of the present study was to report the preliminary results at the five-year follow-up of a prospective cohort study involving computer-assisted rehabilitation of atrophic and resected maxillae via zygomatic implant-supported prostheses and to compare the clinical outcomes between oncologic patients and patients with atrophic maxillae. The primary objective was to assess the zygomatic implant success, survival, and complication rates. The secondary objective was to determine the prosthetic success, survival, and complication rates. Change in the quality of life after rehabilitation and the positional difference between the placed implants and the planned implants were also evaluated.

## 2. Materials and Methods

Patients indicated for zygomatic implant-supported rehabilitation at the Oral and Maxillofacial Surgery Unit of the University of Bologna were identified from October 2013 to January 2019. The inception cohort was defined according to the following inclusion and exclusion criteria.

Inclusion criteria:

Age over 60 years

Need for maxillary rehabilitation

Lack of bone height in the maxillary posterior region due to pneumatization of the sinus and/or resection for cancer

Standard implants are impossible to place in the anterior maxillary region due to the lack of height and thickness of the alveolar ridge consequent to bone resorption and/or resection for cancer

Contraindication or refusal for bone grafting

Failure of previous reconstructive procedures

Good general prognosis

Exclusion criteria:

Opportunity of an alternative treatment with bone grafting and standard implants

Unacceptable standards of health for undergoing the proposed surgical procedures.

Two incident cohorts were created: a cohort of patients who underwent maxillectomy for oncologic reasons (oncologic group) and a parallel cohort of patients with maxillary atrophy (atrophic group). The study began with the placement of zygomatic implants. Censoring criteria included patients’ death, withdrawals, and loss to follow-up.

The cohorts were matched in terms of gender and age distribution.

The hospital’s institutional review board approved this study protocol and ethical approval was granted by the Ethics Committee of Bologna (093\2013\O\SPER). All participants provided written informed consent and the study was carried out in accordance with the Declaration of Helsinki and the European Standards for Good Clinical Practice.

The Strengthening the Reporting of Observational Studies in Epidemiology (STROBE) statement checklist was used for reporting.

### 2.1. Preoperative Procedure

Each patient recruited in the study underwent a general health evaluation and the clinical diagnosis of advanced maxillary bone defect was confirmed according to the inclusion criteria. The pre-existing mobile prosthesis was evaluated for acceptable functional, phonetic, and esthetic characteristics. In case of inadequacy, the prosthesis was modified or a new prosthesis was delivered at least 1 month before the surgery. A new prosthesis was delivered also in cases where the patient had no current prosthesis.

A duplicate of the prosthesis was manufactured in acrylic resin and evaluated intraorally. After approval by the prosthodontist, the prosthesis was made radiopaque on the external surface by using laboratory adhesive mixed with barium sulfate powder or by insertion of four radiographic gutta-percha markers at the canine and the second premolar positions. The duplicate prosthesis was used as a radiographic template for preoperative radiographic surgical planning. The same prosthesis was used in the prosthetic phase after the implant placement as a tray for the impression, allowing the reproduction of the occlusal relationship.

After panoramic radiography and teleradiographic (lateral and frontal) examinations were performed, each patient underwent a preoperative cone-beam computed tomography (CBCT) examination of the maxillofacial region wearing the radiographic template and a marker plate fixed on a provisional implant previously placed in the residual maxillary bone. This marker plate contained fiducial markers essential for the calibration of the dynamic navigation system (ImplaNav, BresMedical, Sydney, Australia) used to guide the zygomatic implant positioning.

### 2.2. Surgical Planning

A surgical plan for each patient was formulated based on the CBCT images using the navigation system software for three-dimensional planning. An anatomically and prosthetically driven approach was utilized to achieve an acceptable surgical and prosthetic outcome according to the zygoma anatomy guided approach [19] for patients in the atrophic group and according to the residual anatomy after resection [16] for patients in the oncologic group. The planned implant position was as external as possible to reduce sinus involvement.

### 2.3. Surgical Procedure

Following general anesthesia and local infiltration of articaine with adrenaline (1:100.000), a mid-crestal incision was made at the level of the alveolar ridge keratinized mucosa with two oblique distal incisions at the level of the maxillary tuberosity. A mucoperiosteal flap was raised to expose the alveolar crest, lateral maxillary wall, pyriform aperture, infraorbital foramen, lateral orbital border, and zygomatic bone. A series of drills mounted on a straight or an angled handpiece (Figure 1a) and an implant motor (Surgysonic^®^ Moto, Esacrom, Imola, Italy) were used to penetrate the alveolar process and the zygomatic bone according to the proposed implant diameter. In some cases, the initial steps were performed using an ultrasonic instrument. The zygomatic implant trajectory was verified in real time by using the ImplaNav dynamic navigation system (Figure 1b) with an intraoral reference tool.

A maxillofacial type navigation system involving the use of an extraoral reference tool (onto the cranial bone) was employed in three oncologic patients in whom the residual maxillary bone was not sufficient for the fixing of the intraoral reference tool. After a final evaluation carried out by a depth gauge, zygomatic implants (Southern Implants, Irene, South Africa) were positioned via the use of implant motor or manually using a fixture mount. Standard zygomatic implants were placed in patients from the atrophic group and in oncologic patients when the implant collar was surrounded by the residual crestal bone (Figure 2a). The oncologic type of zygomatic implant was placed when there was only soft tissue around the implant collar (Figure 2b).

All oncologic patients underwent an oral-antral/nasal primary closure at least with soft tissue. The conical abutment was screwed onto each implant at a torque of 25 NCm using a dynamometric wrench. Finally, the impression transfers were positioned and the flap was sutured (Vicryl 4-0, Ethicon, Somerville, MA, USA).

### 2.4. Prosthetic Procedure

After the flap closure, a sterilized rubber dam perforated at the position of the already mounted transfers was set up. The resin template that was already used for the CBCT was drilled on the palatal side to allow the placement of the transfers. The screw-retained transfers were splinted together using inlay pattern resin (Duralay, Reliance, Worth, TX, USA) The resin template was also used as an impression tray. The impression was made using polyether material injected through the free spaces using a carrier syringe. Very short transfers were chosen to keep the template (reproducing the previously evaluated prosthesis) in occlusion during the setting of the impression material. After transfer removal, healing abutments were positioned onto the implant connections and the impression tray was sent to the technician. After 48 h, a metal framework with the front teeth and the occlusal plate in position was evaluated intraorally. After 72 h, the final provisional screw-retained prosthesis was evaluated intraorally and delivered. In some cases, where the framework check was incorrect, the prosthesis delivery was delayed by approximately 120 h. In addition, radiographic evaluation was performed after loading (Figure 3a,b).

The prosthesis was kept in situ without removing it for at least 3 months. It was manufactured with the freeway space to enable proper oral hygiene due to the frequent palatal emergence of the implant abutment. The present protocol allowed the delivery of a definitive prosthesis after a minimum duration of 3 months in cases of new prosthesis fabrication as well as in cases where the fixed prosthesis needed to be converted into an implant-supported prosthesis.

### 2.5. Clinical and Radiological Follow-Up

Clinical (Figure 4) and radiological follow-up visits were scheduled at 1, 6, 12, 18, 24, 36, 48, and 60 months after the implant placement.

Radiologic examination included an orthopantomographic examination (Figure 5a,b) every year and a lateral teleradiographic examination after implant loading. A CBCT examination was performed at 1 year after the surgery and was repeated if needed. All patients underwent a strict professional hygiene maintenance protocol with 3-month recalls. Instructions and devices for maintaining proper oral hygiene at home were provided to the patients.

### 2.6. Outcome Definition and Assessment

The primary outcome was implant success. The secondary outcomes were implant survival, prosthetic survival and success, change in the quality of life after rehabilitation, and the difference between the position of the planned implants and the placed implants.

### 2.7. Survival Rates and Clinical Success Rates

Implant survival was defined as the absence of failure and the maintenance of the implant function. In case of implant failure during the follow-up, the failure time was recorded and the event was considered a censored event.

The criteria for implant clinical success were fulfilled when the implant remained intact in situ without pain, rotation, bleeding/pus, or soft tissue inflammation. Every incidence of inflammation that did not undergo complete regression with oral hygiene and application of a local disinfectant was classified as unsuccessful.

The prosthetic survival was established by the maintenance of the prosthesis function. Prosthesis was considered successful when it remained in situ with the support of all the placed implants with the denture extension up to the first molar and when there was no need for the prosthesis extension to be reduced from the day of delivery to the final follow-up.

All minor or major complications and adverse events were recorded intraoperatively as well as during the entire follow-up period by a blind operator. All biological and mechanical complications were considered and noted. Additionally, each defined radiographic sign of pathology was recorded.

### 2.8. Comparison between Planned and Real Implant Lengths

The planned implant lengths were compared with the real implant lengths and the difference was measured.

### 2.9. Oral Health Impact Profile (OHIP-14)

The oral health-related quality of life in edentulous adults was assessed using the OHIP-14 questionnaire [22]. The satisfaction level and the masticatory capacity were recorded according to the ability or the lack of ability to bite hard and soft foods and according to the perception of satisfaction related to esthetics, pleasure, comfort, and self-assurance during oral functions. The OHIP-14 questionnaire was administered by a blind operator preoperatively and postoperatively at 1, 6, 12, 18, 24, 36, 48, and 60 months after positioning the zygomatic implants. Patients answered 14 questions with the answer scale ranging from 0 to 4 [22,23]. The highest scores represented the worst satisfaction levels and the lowest scores represented the best satisfaction levels. The maximum possible score was 56. Results were translated into percentage values of satisfaction: 0% represented the worst possible satisfaction level, while 100% represented the best possible satisfaction level. The satisfaction score was assessed at every established follow-up (t). Subsequently, the variation between the follow-ups (Δt) and the difference between the initial and the final satisfaction levels (Δt0–tf) were calculated.

### 2.10. Statistical Analysis

A descriptive analysis was performed that presented the continuous variables as mean ± standard deviation or median ± interquartile range and the categorical variables as absolute and relative frequencies. A survival analysis was carried out using Kaplan-Meier survival curves to study the implant and prosthetic survival and the success over time by analyzing the time to the first event. The log-rank test and Wilcoxon test were used to assess the equality of survival functions across the groups. Adjusted survival curves and hazard functions were also estimated. A Cox regression model was used to analyze the implant and prosthetic risk of failure and unsuccessful events associated with the bone defect etiology (oncologic or atrophic). A variance-corrected Cox proportional hazard model with an Andersen-Gill approach was used to analyze repeated complications for a single implant or prosthesis. The models were adjusted for potential confounders such as age, sex, smoking habit, radiotherapy, and implant type. In addition to the log-linearity of the hazard function for the continuous predictors, the proportional hazard function validity was verified using the Shoenfeld test and scaled Shoenfeld residuals. The lack of interactions between the predictors was also checked. Differences in mean OHIP variations between the atrophic and the oncologic groups were tested with a two-sided t-test after the verification of the normality assumption with the Shapiro-Wilk test. Due to the non-normal distribution of the data, the Wilcoxon rank sum test was performed to analyze the differences between the planned and the placed implants in both the groups. The significance level was set at 0.05. All analyses were performed using Stata version 15 (Stata Corp. LP, College Station, TX, USA).

### 2.11. Sample Size Calculation

The sample size was estimated considering an alpha error of 0.05, power of 0.80, an implant success probability of 0.85 for the atrophic group, and a hazard ratio (HR) of 3.5.

## 3. Results

Twenty patients were recruited including ten with a maxillary bone defect due to resection for cancer and ten with severe maxillary atrophy. The mean age was 64.9 ± 11.5 in the atrophic group and 66.5 ± 13.6 in the oncologic group.

In the oncological group, nine patients (90%) were treated for oral squamous cell carcinoma. One patient had maxillary resection for adenoid cystic carcinoma of minor salivary glands. Regarding the surgical resection, four patients (40%) had maxillectomy type Ib according to Brown classification [16]; one patient (10%) had maxillectomy class Id; three patients (30%) had maxillectomy type IIb; and two patients (20%) had maxillectomy type IIIb. All oncological patients had soft tissue reconstruction performed using local or revascularized flaps. The surgical defect was restored using the temporalis muscle flap (7 patients) and anterolateral thigh free flap (3 patients).

In the oncologic group, four (40%) patients underwent post-operative radiotherapy on the surgical field. The average dose of radiation was 60 Gy (range: 59–63 Gy). The average latency period between the end of treatment and insertion of the implants was 18 months (range: 14–24 months).

The characteristics of the patients are summarized in Table 1. Altogether, 73 zygomatic implants were placed including 38 implants (52%) in the atrophic group and 35 implants (48%) in the oncologic group. Seventeen patients (85%) received four zygomatic implants, while two zygomatic implants were placed in three patients (15%). After surgery, the patients were followed up for an average duration of 39.9 ± 19.5 months. One patient was lost to follow-up at 34 months, one patient died at 6 months, and another died at 39 months.

### 3.1. Implant Survival, Success, and Complications

Two implants failed in the first year. One implant failed in each group. The five-year implant survival rates were 97.4% (95% confidence interval (CI): 82.7–99.6%) in the atrophic group and 96.7% (95% CI: 79.2–99.5%) in the oncologic group. The survival functions did not differ significantly (*p* = 0.89).

The one-year probability of biological implant complications was 5.2% (95% CI: 1.3–19.4%) for the atrophic group and 3.2% (95% CI: 0.4–20.7%) for the oncologic group. The five-year probability was 13.1% (95% CI: 3.7–41%) and 39.7% (95% CI: 9.7–91.7%) in the atrophic and in the oncologic group, respectively. The risk of biological implant complications was not associated with the etiology of the bone defect (HR = 0.72; z = −0.24; *p* = 0.81). Adjusted curves representing the cumulative hazard of implant complications over time are shown in Figure 6.

Biological complications, especially transient mucositis, were observed. These complications were resolved with professional oral hygiene and topical application of chlorhexidine. One patient in the atrophic group reported an extraoral swelling episode at the 51-month follow-up and was treated with the administration of antibiotics (amoxicillin). Clinical examination and CBCT showed no signs of sinusitis. The follow-up continued for up to five years without further episodes.

Considering only the events leading to unsuccessful implant conditions, the one-year clinical implant success probability was 97.3% (95% CI: 82.7–99.6%) for the atrophic group and 96.7% (95% CI: 79.2–99.5%) for the oncologic group. The five-year implant success estimate was 89.8% (95% CI: 60.4–97.7%) for the atrophic group and 96.7% (95% CI: 79.2–99.5%) for the oncologic group. The difference was not statistically significant (*p* = 0.82).

No major complications were reported. Minor surgical complications recorded at the patient level included four cases of transient unilateral paresthesia at the level of the upper lip (all in the atrophic group) and one case of postoperative transient anosmia (in the oncologic group). All of these complications regressed spontaneously within six months after support drug therapy with α-Lipoic acid per os.

### 3.2. Prosthetic Survival, Success, and Complications

In all cases, the prosthesis remained in situ. Thus, the five-year prosthetic survival estimate was 100%.

All unsuccessful prosthetic events occurred in the first year and the majority of them were experienced by the oncologic group. The success probability for both groups remained constant over time with values of 90% (95% CI: 47.3–98.5%) for the atrophic group and 58.3% (95% CI: 22.9–82%) for the oncologic group.

Minor prosthetic complications included unscrewing of the prosthesis screw fractures, chipping and abutment fractures.

The one-year prosthetic complication probability was 5.3% (95% CI: 1.34–19.4%) and 3.2% (95% CI: 0.4–20.7%) for the atrophic group and the oncologic group, respectively. The five-year follow-up complication probability values were 10.7% (95% CI: 4.1–26%) and 50.7% (95% CI: 26.9–79.7%), respectively, for the atrophic group and 10.7% (95% CI: 3.5–29.6%) and 26.7% (95% CI: 12.4–51.9%), respectively, for the oncologic group. The functions representing the prosthetic complications over time did not differ significantly between the groups (*p* = 0.77).

All complications were resolved with component replacement and prosthesis restoration. In three patients from the oncologic group, the denture extension had to be reduced till second premolar due to friction with the soft tissue scar. The risk of prosthetic complications was not affected by the etiology of bone defect (HR = 1.16, 95% CI: 0.39–3.41, z = 0.28, *p* = 0.78). The adjusted cumulative hazard functions are shown in Figure 7.

### 3.3. Implant Length

The mean differences between the planned implant length and the actual implant length were 1.23 ± 2.06 mm for the atrophic group and 0.92 ± 1.36 mm for the oncologic group. No statistically significant difference was observed between the groups (z = 0.30, *p* = 0.76).

### 3.4. OHIP-14

The mean OHIP scores at t0 were 31.4 ± 11.2 for the atrophic group and 29.2 ± 7.6 for the oncologic group. At tf, the scores were 9 ± 1 and 6.3 ± 3.8, respectively, for the two groups. The mean (± sd) OHIP scores are summarized in Table 2. Thus, the satisfaction rates at t0 and tf were 37.2% and 83.9%, respectively, for the atrophic group and 47.9% and 88.8%, respectively, for the oncologic group.

Considering the sum of score variations from the preoperative scores to the scores at the last follow-up, the OHIP-14 score decreased by 71.3% in the atrophic group and by 82.9% in the oncologic group. The difference between the groups was not statistically significant (t = −0.56, *p* = 0.5).

## 4. Discussion

The results showed an overall five-year implant survival rate of 97.1% and a prosthetic survival rate of 100%. The implant survival rates were 97.4% and 96.7% for the atrophic group and the oncologic group, respectively. These results are comparable to those reported in the literature and slightly higher than those reported previously for oncologic patients. A meta-analysis by Chrcanovic [2] reported a five-year cumulative survival rate of 97.5%. In patients with maxillary atrophy, the mean survival rates reported by other studies were approximately 96% after five years of follow-up [12] and after 40-months of follow-up [24]. For oncologic patients, the reported survival rates were 92% [25], 78.6% [26], 85.7% [27], and 91.7% [28].

The use of three-dimensional technology can simplify the zygomatic implant rehabilitation approach. The virtual planning of the implant allows the surgeon to have a clear preoperative vision of the anatomical structures. Virtual planning also seems to provide a good approximation of the implant length. The mean differences between the planned and the actual implant lengths were 1.23 ± 2.06 mm and 0.92 ± 1.36 mm for the atrophic and the oncologic group, respectively. Thus, the need for a stereolithographic model to simulate the surgery beforehand can be avoided [29]. Real-time surgical navigation can aid the surgeon in reproducing the planned implant trajectories [30]. Several clinical trials have reported satisfactory accuracy of dynamic navigation in standard implant placement [31,32,33]. Only a few studies [30,34] have reported on the dynamic navigation concept for zygomatic implant placement, but with very encouraging results including a deviation of 1.35 mm at the entry point and a deviation of 2.15 mm at the apex [34]. Zygomatic implant site preparation is a demanding technique due to the need of a very long drill. To overcome this issue, several methods have been proposed for better control using ultrasonic instruments with or without real-time tracking of the tip [35,36,37]. Some techniques for static guidance are also reported in the literature, but these require more clinical steps and additional costs for surgical stent preparation and manufacturing. The success rate of the zygomatic implants cannot be evaluated with findings related to standard dental implants [38] due to the unconventional engagement at the level of the bone crest and the different esthetic parameters. The present study considered the clinical implant and prosthetic success, which are defined in the “Materials and Methods” section. Success was defined as the absence of any biological or biomechanical problem that required treatment by the surgeon or the prosthodontist. The present study found a one-year clinical implant success rate of 97.3% (95% CI: 82.7–99.6%) for the atrophic group and 96.7% (95% CI: 79.2–99.5%) for the oncologic group. Although the success probability dropped over time, reaching values of 89.8% (95% CI: 60.4–97.7%) for the atrophic group and 96.7% (95% CI: 79.2–99.5%) for the oncologic group at the five-year follow-up, the difference between the groups was not statistically significant. The success rate was slightly higher than that reported by other studies, probably because these studies considered radiologic findings without symptoms [39]. Davo and Pons [12] reported a complication rate of 50% in a five-year clinical study. The one-year probability of implant complications was 5.2% (95% CI: 1.3–19.4%) for the atrophic group and 3.2% (95% CI: 0.4–20.7%) for the oncologic group. It was increased to 13.1% (95% CI: 3.7–41%) and 39.7% (95% CI: 9.7–91.7%), respectively at the five-year follow-up. Although the risk was not statistically associated with the group (etiology of bone defect), oncologic patients suffered from more biological complications in the long term, perhaps due to the unfavorable anatomic characteristics of the resection site, which were less accessible to hygiene maintenance procedures. The clinical success of the prosthetic was lower in the oncologic group (58.3%) than in the atrophic (90%) group. Scar tissue modifies the functional intraoral freeway space and the morphology of the soft tissue. This often necessitates a compromise with non-ordinary tooth positioning or a reduction in the extension of the prosthesis [25]. Several minor biological complications occurred in both groups, but without significant differences. Soft tissue inflammation was observed in case of plaque deposit. However, it regressed with topical antiseptic therapy alone in cases of early identification. Thus, oral hygiene maintenance and evaluation are mandatory. Sinus membrane inflammation is the most frequent complication reported in the literature [2]. However, no case of sinus inflammation with the osteomeatal complex involvement was observed in the present study. No patient reported any symptoms of sinusitis requiring surgical or pharmacological treatment. Preoperative planning and guided exteriorized positioning probably contributed to the reduction of complications. Several minor biomechanical complications of the prosthesis were observed. Among these, the most frequent were fracture of the abutment screw and chipping of the prosthesis. These complications are less common in standard implant rehabilitation [40]. Other authors have also reported these complications in zygomatic implants [1,2,12]. The zygomatic implant-supported prosthesis endures very high stress. Hence, an implant-supported overdenture could be evaluated as a prosthesis. The difference between the two groups suggests that patients from the atrophic group developed higher forces, probably due to more effective mastication. The quality of life of all patients increased significantly after the rehabilitation with a radical change in function and comfort (Figure 8).

The pre-existing removable denture cannot be stable due to atrophied jaw or lack of bone support due to resection. The data from previous literature is consistent with this finding [11,41]. Rehabilitation with zygomatic implants seems to be a reliable option for patients with maxillary atrophy as well as for oncologic patients. The use of three-dimensional computer technologies can simplify the procedure by increasing the diagnostic power as well as the safety and the accuracy of the procedure. Despite the findings reported in the literature, a longer follow-up period is required to confirm the results. Another limitation of this study that could affect the consistency of the results is the limited sample size. Moreover, the study design inevitably affected the internal validity.

However, the results of the present study suggest that this method is worthy of consideration when bone reconstruction and standard implants are not indicated.

## Figures and Tables

**Figure 1 jcm-09-03254-f001:**
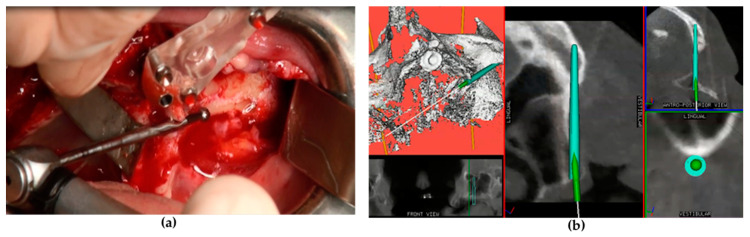
Implant site preparation with the drill mounted on the angled handpiece (**a**) and real-time tracking of the drill via the navigation system screen (**b**).

**Figure 2 jcm-09-03254-f002:**
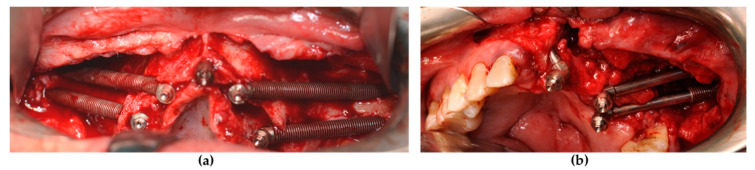
The zygomatic implant was placed and the conical abutment was screwed before suturing. (**a**) The standard type of zygomatic implant was used for patients with maxillary atrophy. (**b**) An oncologic type of zygomatic implant with a machined coronal part was used for implant sites that lacked crestal bone.

**Figure 3 jcm-09-03254-f003:**
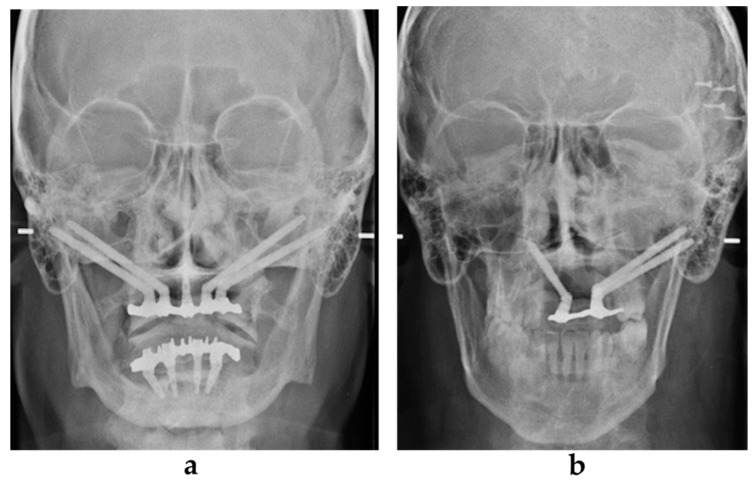
Post-loading radiographic evaluation in a patient with maxillary atrophy (**a**) and in an oncologic patient (**b**).

**Figure 4 jcm-09-03254-f004:**
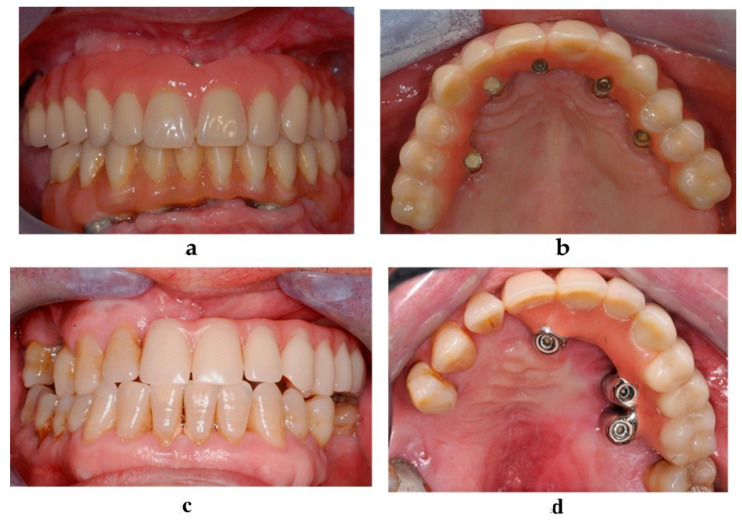
Clinical evaluation and soft tissue maintenance in the atrophic group (**a**,**b**) and in the oncologic group (**c**,**d**): frontal view (**a**), palatal view (**b**).

**Figure 5 jcm-09-03254-f005:**
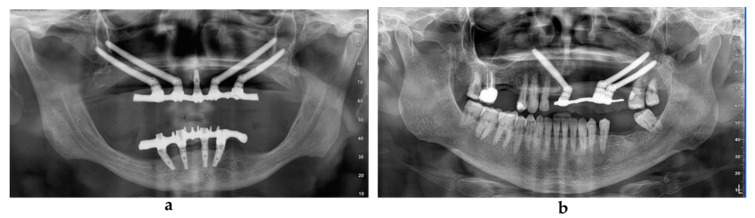
Orthopantomographic evaluation at 4 and 5 years after the surgery in the atrophic group (**a**) and in the oncologic group (**b**).

**Figure 6 jcm-09-03254-f006:**
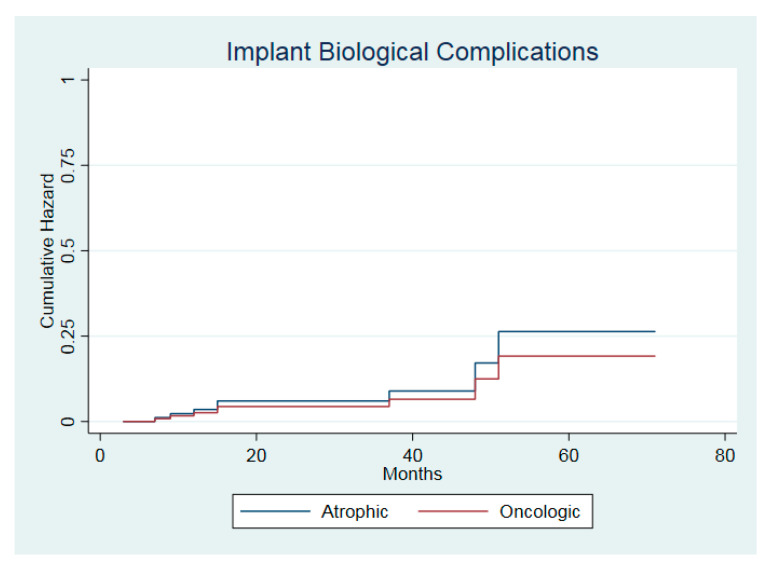
Hazard function of implant biological complications in the two cohorts.

**Figure 7 jcm-09-03254-f007:**
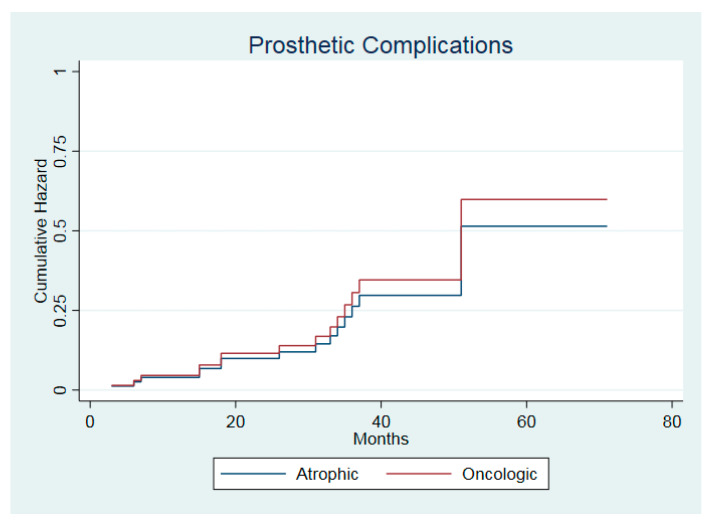
Hazard function of prosthetic complications in the two cohorts.

**Figure 8 jcm-09-03254-f008:**
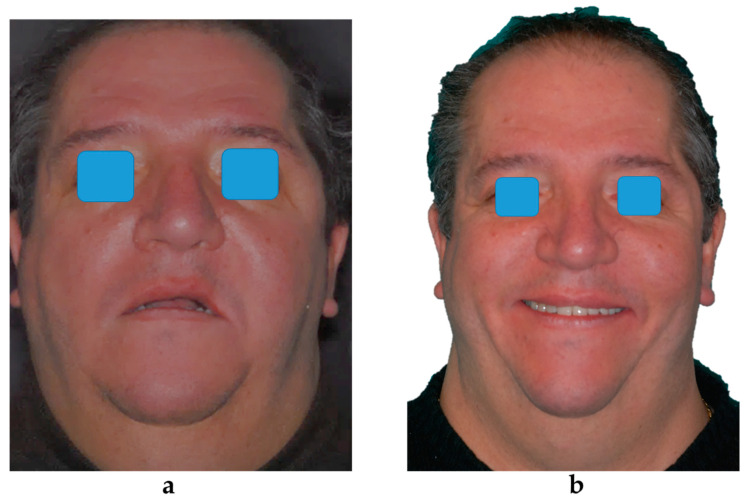
Preoperative (**a**) and postoperative (**b**) facial appearance.

**Table 1 jcm-09-03254-t001:** Patient characteristics.

	Total	Oncologic	Atrophic
Patients (n)	20	10	10
Age (Years)	65.7 (12.2)	65.5 (13.6)	64.9 (11.5)
Males, %	13 (65)	7 (53.8)	6 (46.1)
Smokers, %	9 (45)	4 (44.4)	5 (55.5)
Radiotherapy, %	4 (20)	4 (40)	0
Implants (n)	73	35	38
Hybrid Surface, %	12 (16.4)	12 (34.3)	0
Rough Surface, %	61 (83.5)	23 (65.7)	38 (100)

**Table 2 jcm-09-03254-t002:** Mean (± sd) Oral Health Impact Profile (OHIP) scores at every follow-up.

	t0	t1	t2	t3	t4	t5	t6	tf
Atrophic	31.40 ± 11.28	8.2 ± 3.73	7.66 ± 6.59	6.22 ± 3.49	5.62 ± 4.68	4.83 ± 4.83	7 ± 7.37	9.0 ± 0.10
Oncologic	29.22 ± 7.57	12.12 ± 5.05	10.4 ± 7.86	6.71 ± 10.93	2.33 ± 1.52	10.5 ± 11.30	5.20 ± 7.72	5.0 ± 4.24
Total	30.36 ± 9.51	9.94 ± 4.68	8.64 ± 6.90	6.43 ± 7.37	4.72 ± 4.26	7.66 ± 8.80	6.18 ± 7.20	6.33 ± 3.78

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
