# Peer review of "Computer-Aided Rehabilitation Supported by Zygomatic Implants: A Cohort Study Comparing Atrophic with Oncologic Patients after Five Years of Follow-Up"

_jcm, 2020, doi:10.3390/jcm9103254_

Round 1

Reviewer 1 Report

This is an interesting piece of work looking at the use of zygomatic implants in the management of atrophic and oncology patients....

It looks at several different aspects including navigation surgery and its accuracy for these 20 patients..., zygomatic implant success and survival, patient related outcomes with the use of OHIP scores and complications

It would be useful to classify the type of oncological defects present in this cohort and the types of pathology being treated....the case illustrated shows a very low level alveolar type defect (brown class 1) rather than a larger maxillectomy defect involving the palate. It would be nice to know if all the defects were similar and whether any free tissue flaps were required for defect closure etc..

As this was a secondary type implant rehabilitation (with placement not at the time of cancer resection) - it would be useful to know the time since tumour resection (& radiotherapy)...

the authors should confirm that the 4 cases treated with radiotherapy received treatment before the zygomatic implant surgery...and some idea of dosages received.

The results of this small study have been over-complicated by the use of a large number of varied and complex statistical analyses which cloud the main issues the authors wish to demonstrate. I would advise that some are removed and more straightforward simple analyses presented on biological and prosthetic complications..

Quoted implant survival rates should be "survival estimates" given that the mean follow up was much less than 5 years

re OHIP scores

Hard to expect to see any significance in only 20 patients (10 in each group) unless something drastic goes on between groups.

Only a fraction of the OHIP data is presented 

There should be quite a bit of data as they say they measured OHIP-14 pre-treatment, 1.6.12.18.24.36.48.60 months
Although only 10 patients were in each group this data could be summarised in a table.

Regarding the computerised planning, again this tends to get lost in the paper with only data presented on the difference between planned and actual "implant length" which seems a strange variable to report as it is easy to predict implant length on a basic scan without navigational surgery.

The authors should expand this section to report on the differences between the implant apex position, the implant head position and the trajectory/angulation of the implant....this would give a much better indication as to whether the navigational surgery made any difference at all.

Author Response

thank you for the time you spent to review our paper.

We think that your suggestions helped us to improve the quality of the manuscript.

Reviewer 1:

1- This is an interesting piece of work looking at the use of zygomatic implants in the management of atrophic and oncology patients....

It looks at several different aspects including navigation surgery and its accuracy for these 20 patients..., zygomatic implant success and survival, patient related outcomes with the use of OHIP scores and complications

It would be useful to classify the type of oncological defects present in this cohort and the types of pathology being treated....the case illustrated shows a very low level alveolar type defect (brown class 1) rather than a larger maxillectomy defect involving the palate. It would be nice to know if all the defects were similar and whether any free tissue flaps were required for defect closure etc..

Response:

Regarding the oncological group, nine patients (90%) were treated for oral squamous cell carcinoma. One patient had maxillary resection for adenoid cystic carcinoma of minor salivary glands. Regarding the surgical resection, four patients (40%) had maxillectomy type Ib according to Brown classification [16]; one patient (10%) had maxillectomy class Id; three patients (30%) had maxillectomy type IIb; two patients (20%) had maxillectomy type IIIb. All oncological patients had soft tissues reconstruction performing local or revascularized flaps. The surgical defect was restored using temporalis muscle flap (7 patients) and anterolateral thigh free flap (3 patients).

We included these data on the results section.

2-As this was a secondary type implant rehabilitation (with placement not at the time of cancer resection) - it would be useful to know the time since tumour resection (& radiotherapy)...

Response: In the oncologic group, four (40%) patients underwent post-operative radiotherapy on the surgical field. The average dose of radiation was 60 Gy (range: 59-63 Gy). The average latency period between the end of treatment and the implants insertion was 18 months (range: 14-24 months).

Data were added to the manuscript.

3-the authors should confirm that the 4 cases treated with radiotherapy received treatment before the zygomatic implant surgery...and some idea of dosages received.

Response: In the oncologic group, four (40%) patients underwent post-operative radiotherapy on the surgical field. The average dose of radiation was 60 Gy (range: 59-63 Gy). The average latency period between the end of treatment and the implants insertion was 18 months (range: 14-24 months).

4- The results of this small study have been over-complicated by the use of a large number of varied and complex statistical analyses which cloud the main issues the authors wish to demonstrate. I would advise that some are removed and more straightforward simple analyses presented on biological and prosthetic complications..

Response:Thank you for your observation. We changed the Results section. Regarding the comparison between the risk of failure and unsuccessful events, we used a Cox model because of different follow-up periods, and because of the impossibility to get a good matching between the cohorts, the model was adjusted for the variables thought to be confounders. Otherwise, some confounding factors could have affected the results.

5- Quoted implant survival rates should be "survival estimates" given that the mean follow up was much less than 5 years

Response:Thank you very much for your observation. We modified some sentences in the text.

6- Only a fraction of the OHIP data is presented. There should be quite a bit of data as they say they measured OHIP-14 pre-treatment, 1.6.12.18.24.36.48.60 months
Although only 10 patients were in each group this data could be summarised in a table.

Response: A table summarizing OHIP scores at every follow-up has been added

7- Regarding the computerised planning, again this tends to get lost in the paper with only data presented on the difference between planned and actual "implant length" which seems a strange variable to report as it is easy to predict implant length on a basic scan without navigational surgery.

Response:Yes, It’s right. It was evaluated the contribution of three-dimensional planning with the opportunity to tilt the cross-section scan on the sagittal plane. In our experience, many colleagues accustomed to working on printed radiographs or only using an MPR view of the CT need to simulate the surgery on a stereolithographic model for choosing the implant length. Very often the difficulty to plan exactly the zygomatic implant length leads to bringing in the surgical theatre a lot of implants in order to cover all the implant lengths possible. So we wanted to analyze how the 3d planning can aid the surgeon to overcome these issues. Of course, this is not related to the use of a navigation system.

8- The authors should expand this section to report on the differences between the implant apex position, the implant head position and the trajectory/angulation of the implant....this would give a much better indication as to whether the navigational surgery made any difference at all.

Response:

Thank you for the question. The present manuscript was mainly focused on the presentations of the preliminary results of the clinical outcomes and the complications. The use of navigation was reported only as information. The evaluation of the accuracy obtained comparing the implant planning and the real position of the implants obtained from the CT is the object of a degree thesis still in progress and will be evaluated in another publication. We considered dispersive too much data to be evaluated all in one paper.

Reviewer 2 Report

Thank you for submitting your manuscript. 

This is a well written publication with good information to instruct surgeons and prosthodontists on the challenges of zygomatic implants in oncologic patients. 

The paper clearly shows that the improvement in QOL is similar between oncologic cases and atrophic maxillary cases, although certain challenges seem to be unique to oncologic cases. 

Despite the small case numbers, it is a valuable contribution to the literature, by a team that is well respected for their work on 3D planning. 

Overall it is a well written paper of interest to readers. 

It compares the use of zygomatic implants in the rehabilitation of oncologic cases versus cases with atrophic maxilla. Overall, drawing the comparison doesn't mean much to the clinicians involved in the case of these patients, as both conditions and the needs of patients in each group are vastly different, it is still informative for comparison of complications, prosthetic outcomes and QOL. 

Although zygomatic implants can be very useful in oncologic cases, it still has not acquired widespread use, and this paper will help guide clinicians to the fact that outcomes are good, and comparable in quality and short term survival to their use in atrophic maxilla cases. 

A few modifications may be considered by the authors:

  1. Discussion - page 11, line 390 - '....less habitual....' is not the ideal word for this, maybe 'less common' is a better term.
  2. Discussion - page 11, line 392-early page 12 - here you are discussing the prosthesis over the zygomatic implants, but it is unclear whether you are recommending a implant supported overdenture or a fixed prosthesis, or a hybrid prosthesis. It sounds to me that you are arguing that an overdenture is fine, but this needs to be made clear. An overdenture to me means a removable denture which links onto the implants with magnet or bulbs, instead, I think you are talking about a hybrid prosthesis where there is an implant supported superstructure with acrylic teeth (similar to a denture, but fixed). Please clarify. Considering you also mention here that there are higher masticatory forces in the atrophic maxilla group, did you prefer a different prosthesis between the two groups? 
  3. Discussion - please include more mention of the limitations of the study. Namely, sample size, non randomised comparison of a heterogenous group of patients, short follow up period.

Author Response

- Discussion - page 11, line 390 - '....less habitual....' is not the ideal word for this, maybe 'less common' is a better term.

Response: Thank you. The term was modificated as suggested.

- Discussion - page 11, line 392-early page 12 - here you are discussing the prosthesis over the zygomatic implants, but it is unclear whether you are recommending a implant supported overdenture or a fixed prosthesis, or a hybrid prosthesis. It sounds to me that you are arguing that an overdenture is fine, but this needs to be made clear. An overdenture to me means a removable denture which links onto the implants with magnet or bulbs, instead, I think you are talking about a hybrid prosthesis where there is an implant supported superstructure with acrylic teeth (similar to a denture, but fixed). Please clarify. Considering you also mention here that there are higher masticatory forces in the atrophic maxilla group, did you prefer a different prosthesis between the two groups? 

Response: Thank you for the question. we reported in the M&M section that after 3 months the prosthesis can be substituted with a definitive one. And the protocol approved by the Ethical Committee allowed also to change the prosthesis from a fix to an implant-supported mobile one ( the term mobile was added in the manuscript in order to make it more clear). This was allowed by the protocol for allowing hygiene maintenance. Really in In the present cases, the prosthesis was never converted into a mobile one in the atrophic nor in the oncologic patients. So we consider this an option to evaluate especially in the oncologic patient but avoided discuss this because we have no data to report.

Discussion - please include more mention of the limitations of the study. Namely, sample size, non randomised comparison of a heterogenous group of patients, short follow up period.                                                              Response:Thank you for your observations: we enriched the section of the study limitation, as you suggested.

Round 2

Reviewer 1 Report

Happy with the modifications to the paper.